# IGF-1 and IGFBP-1 as Possible Predictors of Response to Lifestyle Intervention—Results from Randomized Controlled Trials

**DOI:** 10.3390/ijms25126400

**Published:** 2024-06-10

**Authors:** Nina M. T. Meyer, Stefan Kabisch, Ulrike Dambeck, Caroline Honsek, Margrit Kemper, Christiana Gerbracht, Ayman M. Arafat, Andreas L. Birkenfeld, Peter E. H. Schwarz, Jürgen Machann, Martin A. Osterhoff, Martin O. Weickert, Andreas F. H. Pfeiffer

**Affiliations:** 1Department of Endocrinology and Metabolism (Diabetes and Nutritional Medicine), Charité Universitätsmedizin Berlin, 10117 Berlin, Germany; nina.meyer@charite.de (N.M.T.M.);; 2German Center for Diabetes Research (DZD), 85764 Neuherberg, Germany; 3Department of Clinical Nutrition/DZD, German Institute of Human Nutrition Potsdam-Rehbruecke, 14558 Nuthetal, Germany; 4Institute of Diabetes Research and Metabolic Diseases (IDM), Helmholtz Center Munich, Eberhard Karls University of Tübingen, 72076 Tübingen, Germany; 5Department of Internal Medicine IV—Endocrinology, Diabetology, and Nephrology, University Hospital Tübingen, 72076 Tübingen, Germany; 6Department of Diabetes, School of Life Course Science and Medicine, King’s College London, London WC2R 2LS, UK; 7Department for Prevention and Care of Diabetes, Clinic of Medicine III, Faculty of Medicine Carl Gustav Carus, Technische Universität Dresden, 01307 Dresden, Germany; 8Paul Langerhans Institute Dresden, Helmholtz Center Munich, Faculty of Medicine, Technische Universität Dresden, 01307 Dresden, Germany; 9Section on Experimental Radiology, Department of Diagnostic and Interventional Radiology, University Hospital Tübingen, 72076 Tübingen, Germany; 10Warwickshire Institute for the Study of Diabetes, Endocrinology and Metabolism, University Hospitals Coventry and Warwickshire NHS Trust, Coventry CV2 2DX, UK; 11The ARDEN NET Centre, ENETS CoE, University Hospitals Coventry and Warwickshire NHS Trust, Coventry CV2 2DX, UK; 12Centre of Applied Biological & Exercise Sciences (ABES), Faculty of Health & Life Sciences, Coventry University, Coventry CV1 5FB, UK; 13Translational & Experimental Medicine, Division of Biomedical Sciences, Warwick Medical School, University of Warwick, Coventry CV4 7AL, UK

**Keywords:** IGF axis, lifestyle intervention, intervention response, prediabetes

## Abstract

Lifestyle interventions can prevent type 2 diabetes (T2DM). However, some individuals do not experience anticipated improvements despite weight loss. Biomarkers to identify such individuals at early stages are lacking. Insulin-like growth factor 1 (IGF- 1) and Insulin-like growth factor binding protein 1(IGFBP-1) were shown to predict T2DM onset in prediabetes. We assessed whether these markers also predict the success of lifestyle interventions, thereby possibly guiding personalized strategies. We analyzed the fasting serum levels of IGF-1, IGFBP-1, and Insulin-like growth factor binding protein 2 (IGFBP-2) in relation to changes in metabolic and anthropometric parameters, including intrahepatic lipids (IHLs) and visceral adipose tissue (VAT) volume, measured by magnetic resonance imaging (MRI), in 345 participants with a high risk for prediabetes (54% female; aged 36–80 years). Participants were enrolled in three randomized dietary intervention trials and assessed both at baseline and one year post-intervention. Statistical analyses were performed using IBM SPSS Statistics (version 28), and significance was set at *p* < 0.05. Within the 1-year intervention, overall significant improvements were observed. Stratifying individuals by baseline IGF-1 and IGFBP-1 percentiles revealed significant differences: higher IGF-1 levels were associated with more favorable changes compared to lower levels, especially in VAT and IHL. Lower baseline IGFBP-1 levels were associated with greater improvements, especially in IHL and 2 h glucose. Higher bioactive IGF-1 levels might predict better metabolic outcomes following lifestyle interventions in prediabetes, potentially serving as biomarkers for personalized interventions.

## 1. Introduction

Obesity and its metabolic sequelae are increasing worldwide, and they are the primary causes of the most prevalent diseases of industrialized countries linked to metabolic syndrome. Lifestyle approaches aiming to prevent diabetes through moderate weight loss, a healthy diet, and increased physical activity have proven highly successful due to improvements in insulin sensitivity and insulin secretion [1]. However, in these trials, some study participants did not show the expected improvements despite experiencing significant weight loss and a reduction in liver fat [2]. These individuals may require more intense programs aimed at lifestyle factors, or they might profit from early pharmacological interventions. To date, there is a lack of biomarkers for identifying these individuals at early time points.

The insulin-like growth factor (IGF) system has evolutionarily been separated from the insulin system in vertebrates to regulate growth- and food-related metabolism with greater flexibility, but it has remained closely intertwined in the insulin/IGF system [3,4]. This system is closely linked to life expectancy, from worms to mammals, such that a reduced function prolongs life, which probably serves to survive unfavorable periods of famine and shortness [5]. In short-lived mammals, a deficiency of the growth hormone IGF axis induces longevity, while this is less clear in long-lived species. Nevertheless, growth hormone (GH) and IGF-1 deficiency is associated with reduced rates of cancer, type 2 diabetes, and diabetic complications in humans, while increased levels of IGF-1 have been linked to the incidence of some types of cancer [5,6].

However, the IGF system is also associated with regenerative processes, low function predicted sarcopenia [7], cardiovascular risk [8], cognitive dysfunction [9], and type 2 diabetes in epidemiological [10] and randomized controlled studies [11]. This discrepancy is possibly related to the complexity of the insulin/IGF-1 system with its regulation by age, nutrition, and behavioral components resulting from epigenetics against the background of a strong genetic disposition [12].

Although circulating IGF-1 is a hepatokine that is produced in response to pulses of GH release from the pituitary gland, mediating most of its actions, it is closely linked to insulin release through the regulation of IGF-binding proteins (IGFBPs) [13,14]. Circulating levels of IGFBPs determine the bioactivity of IGF-1 by binding 99% of IGF-1 [3,13,15]. IGFBP-1 and IGFBP-2 were shown to be closely related to circulating insulin and insulin sensitivity [13,16,17]. Circulating IGFBP-1 is produced by the liver and suppressed by portal levels of insulin [18]. Acute increases in insulin after meals decrease IGFBP-1 by up to 60% and thereby increase free, biologically active IGF-1 [18,19,20]. Chronically increased levels of insulin decrease circulating IGFBP-1, which correlates closely with whole-body and hepatic insulin resistance as well as with metabolic dysfunction-associated steatotic liver disease (MASLD) [14,18]. IGFBP-2 is also related to insulin resistance but is regulated more slowly by insulin [16]. IGF-1 and the IGF-binding proteins regulate visceral and subcutaneous fat depots and exert significant effects on the hepatic fat content [3,13,14,15,20,21].

The IGF system shows significant inheritance and additionally appears to be regulated by epigenetic factors programmed by obesity, diet, and physical activity, and thus individual metabolic conditions [12,22]. Furthermore, in a preceding study, we found low IGF-1 and high IGFBP-1 levels to be predictive of the incidence of T2DM in a prediabetic cohort with a high risk for the development of diabetes. This phenomenon is most likely attributable to impaired beta cell function, possibly explaining a non-responsiveness to lifestyle interventions [11].

Consequently, we hypothesized that components of the IGF system might be related to the success of metabolic improvements in lifestyle studies and may allow for the prediction of responsiveness to intervention studies. We therefore investigated the role of the IGF system in predicting the success of lifestyle interventions in people with impaired glucose metabolism by analyzing the above-mentioned high-risk prediabetic cohort.

## 2. Results

### 2.1. Baseline Characteristics

Most participants of the studies displayed characteristics of metabolic syndrome, including abdominal obesity and impaired glucose metabolism at baseline (Table 1). At baseline, the absolute levels of IGF-1 showed a wide spread between individuals and were correlated negatively with fasting glucose, waist/hip ratio (WHR), and VAT (measured via MRI), but not with IHL (measured via magnetic resonance spectroscopy, MRS) or indices of insulin sensitivity or insulin secretion, as already reported elsewhere [11]. IGFBP-1 similarly showed a wide variation and not only correlated significantly inversely with constitutional markers such as body mass index (BMI), WHR, VAT, and IHL, but also positively with indices of glucose sensitivity and secretion [11]. IGFBP-2 displayed modest negative correlations with anthropometric markers (BMI, VAT, and IHL) [11].

The differences in metabolic parameters at the baseline between the subgroups of IGF-1 and IGFBP-1 (sub- vs. supra median levels), respectively, are depicted in Appendix A. Here, a similar pattern emerges, as seen in the correlation analyses: high IGF-1 and high IGFBP-1 levels are each associated with more favorable metabolic profiles.

### 2.2. Responses to Lifestyle Interventions

Lifestyle interventions led to highly significant improvements in anthropometric and metabolic parameters of the participants, as already reported elsewhere [11]. However, this was not reflected by major changes in IGF-1 or binding proteins, which showed no changes except for a statistically significant but small increase in IGFBP-1 from 2.1 to 2.2 µg/L, despite having significant correlations with metabolic parameters [11].

We therefore tested whether higher or lower baseline levels of IGF-1 and IGFBP-1 might be associated with responses to lifestyle interventions and compared changes in individuals with levels above or below the medians.

### 2.3. Responses to Lifestyle Interventions within Median Subgroups of IGF-1 and IGFBP-1 Baseline Levels

The stratification of individuals by baseline IGF-1 and IGFBP-1 medians (median subgroups) revealed highly significant differences between the groups in terms of changes in the IGF-1 and IGFBP-1 levels as well as changes in metabolic parameters during the intervention.

Participants with baseline IGF-1 levels above the median showed a highly significant decrease in IGF-1 levels, whereas individuals with baseline levels below the median showed increased IGF-1 levels (Table 2a).

Concerning the IGFBP-1 subgroups, individuals with levels below the median showed a significant increase, whereas individuals with levels above the median showed a significant decrease in IGFBP-1 levels (Table 2b).

Moreover, individuals with lower IGFBP-1 concentrations at baseline exhibited a significant increase in IGF-1 levels during the intervention, whereas those with higher IGFBP-1 levels had significantly decreased IGF-1 levels (Table 2b).

### 2.4. Differential Response to Lifestyle Interventions Depending on Baseline Levels of IGF-1 and IGFBP-1

The differential change in IGF-1 levels between subgroups below and above the median led to a significant between-group difference (Table 3a), which persisted after adjusting for change in BMI (Mean difference (MD) = 32.6 µg/L; 95%-CI: [23.5; 41.7]; F(1, 339) = 49.4; *p* = <0.001; partial η^2^ = 0.127; note: homogeneity of regression slopes not given for interaction term subgroup × change in BMI; homogeneity of variances not given).

Regarding intervention-induced metabolic changes, individuals within the subgroup with supra-median levels of IGF-1 mostly showed improved profiles compared to those with lower IGF-1 levels in comparison to the baseline despite having similar reductions in body weight (Table 2a and Table 3a). Specifically, while both groups experienced significant reductions in VAT and IHL, these reductions were significantly greater in the subgroup with supra-median levels. Regarding glucose metabolism, fasting glucose and 2 h glucose improved significantly in both groups, but fasting insulin and homeostatic model assessment of insulin resistance (HOMA-IR) only improved significantly in the subgroup with supra-median levels. Fasting insulin and the Matsuda index showed significantly greater improvements in this group. When adjusted for age to baseline, significant between-group differences were comparable (∆ IGF-1 and ∆ IHL), attenuated (∆ VAT), or diminished (∆ fasting insulin and ∆ Matsuda).

Furthermore, changes in the IGFBP-1 levels showed significant between-group differences when comparing the two subgroups of IGFBP-1 baseline levels (Table 3b).

Both IGFBP-1 subgroups showed overall metabolic improvements upon intervention with regard to BMI, total and visceral fat volumes, IHL, fasting, and 2 h glucose as well as fasting insulin levels, insulin sensitivity, and insulin secretion (Table 2b). Except for the fasting glucose levels, each of these improvements were more pronounced within the group with lower IGFBP-1 levels at baseline. However, significantly greater improvements were only observed for changes in IHL (Table 3b).

Sensitivity analyses performed by dividing the cohort into tertiles according to the IGF-1 and IGFBP-1 baseline levels (see Appendix A) revealed that significant associations persist, or in the case of IGFBP-1, may even be strengthened or newly emerged (e.g., a greater decline in BMI in the group with low IGFBP-1 compared to the group with higher values).

## 3. Discussion

It is well established that IGF-1 and IGFBP-1 are highly heritable and correlate with anthropometric and metabolic parameters beyond inheritance [22,23]. Here, we show that the responses of IGF-1 and IGFBP-1 to lifestyle interventions depended on the baseline expression levels. Moreover, the baseline levels predicted the ability to respond to lifestyle changes and thereby appear to determine the success of lifestyle interventions.

The baseline levels of IGF-1 vary widely between individuals, primarily due to inheritance [12,24] and due to the parameters of glucose and insulin metabolism [23]. Although caloric and primarily protein restriction reduce IGF-1 [25], previous studies did not observe significant changes in IGF-1 upon lifestyle interventions and weight loss at 1 or 6 years [26], nor did they report a decrease in IGF-1 [27]. Unexpectedly, upon moderate weight loss, we observed highly significant increases in IGF-1 in people with low levels at baseline, while IGF-1 decreased in participants with initially high levels. Due to the wide spread of baseline levels, the absolute values were still lower in the subgroup with sub-median levels and higher in the group with supra-median levels after the intervention (Figure A1); this was possibly due to the strong inheritance, which was estimated at 63% in twin studies [12,24]. Higher levels of IGF-1 were associated with a reduced risk of type 2 diabetes in cross-sectional [10] and prospective [28,29] epidemiological studies, but they were also associated with an increased risk in a Mendelian randomization study [30].

The observed changes in the IGF-1 and IGFBP-1 levels were relatively small. The subgroups with levels above and below the median were closely clustered together and significantly above zero, excluding the possibility of a floor or ceiling effect. Despite this proximity of the subgroups, there was an observable tendency for high IGF-1 levels trending downward and low IGF-1 levels trending upward. A similar pattern was observed with IGFBP-1. This may potentially represent a simple regression to the mean. However, the contrary argues that the split between the subgroups below and above median levels was associated with significant metabolic consequences, which is an intriguing and novel aspect.

Our data show that higher levels of IGF-1 were predisposed to significantly greater improvements in intrahepatic lipids and in the visceral fat volume, markers which are strongly associated with metabolic syndrome, insulin resistance, and diabetes risk despite comparable weight loss. In addition, fasting insulin only decreased in individuals with higher IGF-1 levels, indicating that the group with low levels was unable to improve insulin sensitivity despite experiencing weight loss and significant reductions in VAT and IHL. IGF-1 may thus have determined the capacity for metabolic recompensation in this high-risk group. Although the levels of IGF-1 are primarily determined by inheritance, protein intake increases IGF-1, while other foods have minor effects. We monitored food intake and did not observe food-dependent effects on IGF-1 in our study, which did not specifically involve high protein intake.

Given that our study was carried out in prediabetic cohorts with a higher risk for progression, it may not be translatable to people without metabolic impairments. However, higher levels of IGF-1 at baseline were also associated with a reduced risk of developing T2DM in our study [11], supporting a protective effect of IGF-1 when undergoing lifestyle intervention.

As a limitation of this study, it should be noted that age may possibly confound the association between IGF-1 levels and metabolic response; it is well known that IGF-1 decreases with age. Older individuals might display a weaker allostatic response, leading to less pronounced changes in response to dietary interventions. Accordingly, some significant differences in metabolic changes associated with IGF-1 levels remained, but others were attenuated or diminished after adjusting for age. This suggests that age is a significant factor in the variability of IGF-1 responses among different cohorts.

In earlier studies, higher levels of IGFBP-1 are generally associated with better insulin sensitivity and insulin secretion, while low levels are prospectively associated with T2DM and IGT [15,17]. IGFBP-1 is acutely and chronically inhibited by portal insulin levels and, therefore, low levels closely reflect the hepatic fat content and hepatic insulin resistance [18]. In our cohorts, we observed similar inverse associations of IGFBP-1 with IHL, VAT, hepatic, and whole-body insulin resistance, reflecting extensive metabolic impairment. It might therefore seem unexpected that low IGFBP-1 levels are associated with considerably greater improvements in anthropometric and metabolic responses to lifestyle interventions despite similar reductions in body weight being achieved. One may argue that greater improvements were due to the greater initial impairments, but higher IGFBP-1 also labeled a group with reduced capacity for improvement. This phenomenon was also observed in earlier studies on individuals with prediabetes, which showed that patients with combined IFG-IGT—a prediabetes subtype with the most prominent alterations throughout the entire metabolism—respond more effectively to lifestyle interventions than individuals with isolated IGT [31].

In fact, the prediabetic group differed from the high-risk groups identified in cross-sectional or prospective observational studies with regard to IGFBP-1; according to a Swedish study, an increase in IGFBP-1 was observed in individuals with prediabetes as they approached overt type 2 diabetes [19,20]. This appears to relate to the progression of hepatic insulin resistance, which reduces the suppression of the hepatokine IGFBP-1 relative to circulating insulin levels [21]. Furthermore, the progressive beta cell dysfunction reflected by impaired glucose tolerance appears to contribute to this phenotype. Notably, in our study, participants with higher IGFBP-1 showed—although statistically non-significant—smaller reductions in the 2 h glucose values and only one-quarter of the reduction in fasting insulin compared to the subgroup with sub-median levels. Higher IGFBP-1 thus labels the group that is unable to improve beta cell function upon reductions in body weight and visceral and hepatic fat contents. Accordingly, high IGFBP-1 levels were also shown to identify people with prediabetes who are unresponsive to standard lifestyle interventions [11].

Mechanistically, this phenomenon described above may be attributed to the antagonism of IGF-1 activity by IGFBP-1, which is particularly pronounced in the interstitial and pericellular environment [13,15,21]. IGF-1 was shown to cooperate with insulin in maintaining beta cell function in adult animals, while its developmental function was negligible [32,33]. The selective deletion of beta cell IGF receptors primarily led to impaired glucose sensing rather than a loss of beta cell mass in mice [32]. This appears to translate to humans, as suggested in our present study, by the protective effects of higher IGF-1 and lower IGFBP-1 levels, leading to increased biologically active IGF-1. In addition, our findings indicate that higher activity within the IGF system appears to support—in the context of intervention—the loss of ectopic fat stores, as shown by the greater reductions in visceral and hepatic fat in this study.

In mice with diet-induced obesity, the overexpression of IGFBP-1 improved insulin sensitivity [34]. In humans, weight loss, reductions in hepatic fat and hepatic insulin resistance and, consequently, reductions in circulating insulin are associated with increases in IGFBP-1 [35], which we also observed in our study in patients with low IGFBP-1 levels at baseline.

Taken together, the IGF-1 system in metabolism represents a complex interplay that certainly requires further investigation. Our novel findings suggest that IGF-1 and IGFBP-1 may serve as serological biomarkers to predict lifestyle responses—which, to our knowledge, would be the first of their kind.

## 4. Materials and Methods

### 4.1. Project Design and Participants

For the analysis, we used data from three German lifestyle intervention studies: the Prediabetes Lifestyle Intervention Study (PLIS), the Diabetes Nutrition Algorithm-Prediabetes Trial (DiNA-P), and the concluded Optimal Fiber Trial (OptiFiT). All three studies focus on lifestyle interventions for individuals with prediabetes at a high risk of developing type 2 diabetes. High-risk criteria included reduced insulin sensitivity together with the presence of MASLD and/or reduced insulin secretion (PLIS, DiNA-P) or impaired glucose tolerance (OptiFiT).

Data for this analysis cover the first year of intervention of all three studies.

PLIS, a multicenter study initiated in 2013 at eight sites in Germany, is part of the national research association, the German Center for Diabetes Research (DZD) [2]. DiNA-P, designed in parallel with PLIS, was intended to offer equivalent data on an alternative dietary intervention and constitutes an independent trial (refer to clinicaltrials.gov: NCT02609243). Our present analysis includes 135 PLIS participants from the University Hospital Carl Gustav Carus of the Technical University Dresden and 116 DiNA-P participants from sites in Nuthetal and Berlin.

The OptiFiT study was conducted between March 2010 and October 2014 in Berlin and Nuthetal [36]. Our analysis included data from 94 participants who completed the first year of intervention.

The primary goal of each study was metabolic improvement and moderate weight loss through lifestyle modification. We assessed changes after a one-year intervention period for each study. Thus, the ultimate cohort comprises 345 participants, from whom the fasting levels of IGF-1, IGFBP-1, and IGFBP-2 at both baseline and after 12 months were collectively available.

### 4.2. Interventions

In PLIS and DiNA-P, participants followed a hypo- to isocaloric diet based on low fat intake, as per the 2018 recommendations by the German Nutrition Society (<30 kcal% fat, <10 kcal% saturated fatty acids, >15 g/1000 kcal fiber/day) for 12 months. They received personalized dietary counseling in 8 or 16 sessions of equal duration depending on randomization. At DiNA-P, there was an additional three-week comparison between reduced carbohydrate or fat intake while maintaining similarity to the PLI study (refer to clinicaltrials.gov; NCT02609243). In both trials, long-term follow-up extended beyond the initial 1-year intervention.

The OptiFiT study focused on insoluble cereal fiber intake’s effects on glycemic metabolism in individuals with impaired glucose tolerance (IGT). Participants underwent random assignment to either cereal fiber or placebo supplementation for a duration of 2 years. Both groups engaged in a structured 1-year lifestyle program adapted from the PREvention of DIAbetes Self-management (PREDIAS, [37]). Details of the study design are published elsewhere [36].

Nutrient and energy intake were monitored via dietary records throughout the studies. All participants were mandated to achieve a certain level of daily physical activity, which was monitored through a combination of questionnaires and technical devices.

Ethical committees approved the study protocols for all three trials, which also adhered to Good Clinical Practice principles and the Declaration of Helsinki. Before enrollment, all participants provided written informed consent and underwent comprehensive medical evaluations, including history, physical exams, and routine blood and urine tests. At the study’s outset, participants had no evidence of severe chronic diseases, including metabolic, cardiovascular, lung, gastrointestinal, and autoimmune diseases or cancer.

### 4.3. Sample Collection and Anthropometric and Metabolic Assessments

In each study, the participants underwent a baseline assessment, which included medical examinations, fasting blood draws, an oral glucose tolerance test (oGTT), anthropometric measurements, and magnetic resonance (MR) examination, along with the provision of food records and activity meters. These assessments were repeated 1 year after enrollment into the respective study. Notably, within the OptiFiT cohort under analysis, MR examination was only carried out on 16 participants.

Measurements of body weights, heights, and circumferences were taken with participants wearing light clothing and no shoes. Fat volumes were assessed using magnetic resonance imaging (MRI), while hepatic fat storage was detected using MR spectroscopy (^1^H-MRS) following a previously published protocol [38]. MR scans were evaluated in a blinded manner by a medical physicist (JM).

Both fasting blood samples and oral glucose tolerance tests (oGTTs) using 75 g of glucose provided the basis for the determination of glucose homeostasis, insulin sensitivity, and insulin secretion. In the PLIS and DiNA-P, blood samples after glucose load were collected at 0, 30, 60, 90, and 120 min. In OptiFiT, capillary blood for the determination of glucose levels and whole blood for insulin measurements were drawn at 0, 60, and 120 min after glucose load, respectively. Acquired blood samples were either analyzed immediately or stored at −80 °C.

We used HOMA-IR and the Matsuda index [39,40] as standard surrogate parameters for insulin resistance (IR). The hepatic insulin resistance (HIRI) was estimated using a formula developed by Abdul-Ghani et al. [41]. Insulin secretion capacity was approximated using the modified insulinogenic index (IGI) according to Seltzer [42] and the disposition index-2 (DI, [43]). For the calculation of oGTT-based indices, only participants with complete data sets for respective required timepoints were analyzed.

### 4.4. Laboratory Analyses

Glucose and insulin levels, along with routine laboratory safety parameters, were measured using established standard methods (for insulin, ELISA by Mercodia^®^, Uppsala, Sweden was used in DiNA-P and OptiFiT an chemiluminescent immunoassay by Siemens Healthcare GmbH, Erlangen, Bavaria, Germany, in PLIS).

For the measurement of fasting levels of IGF-1, IGFBP-1, and IGFBP-2, we used commercially available ELISA assays (Mediagnost^®^, Reutlingen, Baden-Württemberg, Germany), which were previously validated by our research group [44], following the manufacturer’s instructions (intra- and interassay coefficients of variation; IGF-1: 5.8% and 8.6%, IGFBP-1: 6.5% and 6.1, IGFBP-2: both <10%). The measurement was performed by technical assistants in a blinded manner.

### 4.5. Statistics

We analyzed the data of 345 participants that had 1 year of follow-up data available.

The data are presented as the mean with the standard deviation (SD) or as the median with the interquartile range (IQR) depending on the distribution of the data.

Within-group differences were assessed using Student’s paired *t*-test (one-tailed) in the case of normal distribution, and Wilcoxon Signed-Rank test was used in the case of skewed data.

Between-group differences were evaluated using Welch’s test (one-tailed testing) in the case of normally distributed parameters regardless of homogeneity of variance, by following a recommendation by Rasch, Kubinger, and Moder [45]. Differences between groups of non-normally distributed data were tested via Mann–Whitney U test. Mean difference (MD) between groups was indicated as mean difference between the subgroup below the median and the subgroup above the median.

We used ANCOVA models to test for between-group differences between two independent groups when we controlled for one or more variables. We used Bonferroni correction to adjust for multiple comparisons. We assessed the homogeneity of regression slopes by testing the interaction terms between covariates and the group variable. We indicated if the analysis must be considered with caution. Using Levene’s test (based on median), we assessed the homogeneity of variances. If these were not given, we acknowledged it but assumed the robustness of ANCOVA models due to roughly equal group sizes.

A two-sided *p*-value of <0.05 was considered statistically significant. The analyses were conducted using IBM^®^ SPSS^®^, Version 28 (SPSS Inc, Chicago, IL, USA).

## 5. Conclusions

In conclusion, our study proposes that the baseline expression levels of IGF-1 and IGFBP-1 play a role in determining the responses to lifestyle interventions, with higher levels of IGF-1 predisposing individuals to greater impairment at baseline, but also greater interventional improvements in metabolic risk markers, despite experiencing similar weight loss. Conversely, low levels of IGFBP-1 are associated with greater improvements in response to lifestyle interventions. These associations are seen in individuals with pre-existing impairments in glucose metabolism. Mechanistically, the antagonistic relationship between IGF-1 and IGFBP-1 might form the basis for these associations.

Understanding these relationships might help to identify individuals who may require more intensive interventions early on. As our data’s applicability is limited to prediabetic high-risk groups, further research is warranted to validate these findings in broader populations.

## Figures and Tables

**Table 1 ijms-25-06400-t001:** The baseline characteristics of the cohort.

Parameters	Value	No.
Women (%)	54.0	186
Age (years)	62.7 ± 8.7	345
Study allocation		
PLIS (%)	39.1	135
DiNA-P (%)	33.6	116
OptiFiT (%)	27.2	94
IGF-1 (µg/L)	141.8 ± 53.7	345
IGFBP-1 (µg/L)	2.1 [1.4; 4.1]	345
IGFBP-2 (µg/L)	259.1 [134.2; 422.6]	345
BMI (kg/m^2^)	30.9 ± 5.4	345
Present overweight (%)	38.0	132
Present obesity (%)	50.7	175
Grade I (%)	29.3	101
Grade II (%)	15.1	52
Grade III (%)	6.4	22
WHR (cm/cm)	0.93 ± 0.09	341
Body fat content_-BIA_ [%]	34.7 ± 8.5	312
VAT_-MRI_ (l)	5.5 ± 2.4	225
IHL_-MRS_ (%-abs.)	7.0 [3.0; 14.4]	231
Present MASLD (%)	39.4	136
Fasting glucose (mmol/L)	5.7 ± 0.7	345
2 h glucose (mmol/L)	8.2 ± 1.6	345
Fasting insulin (pmol/L)	73.4 [51.7; 105.5]	337
Present IFG + NGT (%)	31.9	110
Present NFG + IGT (%)	31.6	109
Present IFG + IGT (%)	36.5	126
HOMA-IR	2.6 [1.7; 3.8]	337
Matsuda Index	2.6 [1.8; 3.5]	238
HIRI	37.2 [30.6; 44.4]	242
IGI	11.7 [7.5; 21.2]	242
DI	30.9 [21.6; 43.6]	238

The data are shown as the mean ± SD (normally distributed variables), as the median [IQR] (non-normally distributed variables), or as proportions (%). PLIS: Prediabetes Lifestyle Intervention Study. DiNA-P: Diabetes Nutrition Algorithm-Prediabetes. OptiFiT: Optimal Fibre Trial. BMI: body mass index. WHR: waist/hip ratio. VAT: visceral adipose tissue. BIA: bioelectrical impedance analysis. MRI: magnetic resonance imaging. IHL: intrahepatic lipid content. MRS: magnetic resonance spectroscopy. abs: absolute. MASLD: metabolic dysfunction-associated steatotic liver disease. IFG: impaired fasting glucose. NGT: normal glucose tolerance. IGT: impaired glucose tolerance. HOMA: homeostatic model assessment. IR: insulin resistance. HIRI: hepatic insulin resistance index (Abdul-Ghani). IGI: insulinogenic index (Seltzer). DI: disposition index-2. IGFBP1/-2: insulin-like growth factor binding protein-1/-2.

**Table 2 ijms-25-06400-t002:** IGF-1, IGFBP-1, IGFBP-1, and metabolic parameters at baseline and after 1 year, respectively, in association with (**a**) IGF-1 baseline levels and (**b**) IGFBP-1 baseline levels.

Parameters	Baseline	1 Year	*n*	*p*	*d*/*r*	Baseline	1 Year	*n*	*p*	*d*/*r*
	(**a**)									
	IGF-1 < 134.2 µg/L					IGF-1 ≥ 134.2 µg/L				
IGF-1 [µg/L]	99.9 ± 23.3	117.2 ± 38.8	172	**<0.001**	−0.56	183.5 ± 41.5	168.7 ± 51.0	173	**<0.001**	0.29
IGFBP-1 [µg/L]	2.2 [1.2; 4.4]	2.5 [1.3; 4.5]	172	0.460	−0.06	2.1 [0.9; 3.7]	1.9 [1.2; 4.0]	173	**0.015**	−0.18
IGFBP-2 [µg/L]	269.6 [148.1; 453.6]	271.7 [162.0; 431.9]	170	0.290	−0.08	251.5 [133.9; 385.2]	250.7 [164.8; 427.7]	172	0.057	−0.14
Body mass index [kg/m^2^]	30.8 ± 5.2	29.9 ± 5.1	171	**<0.001**	0.51	31.1 ± 5.6	29.9 ± 5.4	171	**<0.001**	0.65
Waist-to-hip ratio [cm/cm]	0.94 ± 0.09	0.92 ± 0.09	166	**0.011**	0.18	0.93 ± 0.09	0.93 ± 0.09	166	0.359	0.03
Body fat content_-BIA_ [%]	35.1 ± 8.6	34.0 ± 9.0	145	**<0.001**	0.34	34.2 ± 8.5	33.1 ± 9.1	147	**0.002**	0.25
Visceral fat volume_-MRI_ [L]	5.6 ± 2.5	5.2 ± 2.4	111	**<0.001**	0.43	5.6 ± 2.3	5.0 ± 2.1	86	**<0.001**	0.71
Intrahepatic lipid content_-MRS_ [%-abs.]	7.0 [3.0; 14.7]	4.4 [2.3; 8.9]	113	**<0.001**	−0.41	7.2 [3.0; 14.2]	3.1 [1.1; 7.1]	89	**<0.001**	−0.68
Fasting glucose [mmol/L]	5.8 ± 0.7	5.6 ± 0.8	164	**<0.001**	0.34	5.7 ± 0.7	5.5 ± 0.7	157	**<0.001**	0.31
2 h glucose [mmol/L]	8.3 ± 1.5	7.6 ± 1.9	164	**<0.001**	0.36	8.1 ± 1.6	7.3 ± 2.0	157	**<0.001**	0.46
Fasting insulin [pmol/L]	79.7 [55.8; 108.2]	77.8 [54.9; 111.2]	170	0.239	−0.09	66.0 [49.6; 99.7]	61.5 [44.1;88.3]	165	**<0.001**	−0.34
HOMA-IR	3.0 [1.9; 3.9]	2.7 [1.7; 3.9]	170	0.051	−0.15	2.4 [1.6; 3.7]	2.0 [1.4; 3.1]	164	**<0.001**	−0.36
Matsuda index	2.5 [1.8; 3.3]	2.9 [2.0; 4.3]	122	**<0.001**	−0.34	2.8 [1.9; 3.6]	3.6 [2.5; 5.0]	112	**<0.001**	−0.50
HIRI	37.5 [30.8; 45.3]	34.2 [29.8; 42.0]	127	**0.003**	−0.26	36.7 [30.0; 42.6]	33.5 [27.2; 39.3]	114	**<0.001**	−0.41
IGI	11.7 [7.3; 21.2]	12.4 [7.8; 19.8]	127	0.273	−0.10	11.6 [7.5; 19.2]	11.2 [7.0; 17.0]	114	**0.232**	−0.11
DI	28.2 [19.5; 43.6]	34.3 [21.4; 63.1]	122	**<0.001**	−0.36	33.6 [22.9; 44.5]	38.7 [25.0; 68.0]	112	**<0.001**	−0.31
	(**b**)									
	IGFBP-1 < 2.13 µg/L					IGFBP-1 ≥ 2.13 µg/L				
IGF-1 [µg/L]	141.5 ± 48.5	150.5 ± 52.5	172	**0.002**	−0.23	142.1 ± 58.5	135.6 ± 50.6	173	**0.043**	0.13
IGFBP-1 [µg/L]	1.0 [0.7; 1.5]	1.5 [0.9; 2.2]	172	**<0.001**	−0.53	4.1 [2.8; 6.8]	3.9 [2.3; 5.6]	**173**	**0.045**	−0.15
IGFBP-2 [µg/L]	223.6 [119.5; 369.2]	237.4 [141.2; 352.5]	172	0.080	−0.13	310.2 [175.4; 463.2]	319.5 [190.2; 515.7]	170	0.179	−0.10
Body mass index [kg/m^2^]	31.8 ± 5.0	30.7 ± 4.8	171	**<0.001**	0.68	30.0 ± 5.7	29.1 ± 5.6	171	**<0.001**	0.49
Waist-to-hip ratio [cm/cm]	0.94 ± 0.08	0.93 ± 0.08	165	**0.035**	0.14	0.93 ± 0.10	0.92 ± 0.09	167	0.096	0.10
Body fat content_-BIA_ [%]	35.5 ± 8.1	34.3 ± 8.8	149	**<0.001**	0.35	33.6 ± 9.1	32.7 ± 9.6	143	**0.003**	0.24
Visceral fat volume_-MRI_ [L]	6.0 ± 2.1	5.5 ± 2.1	106	**<0.001**	0.64	5.1 ± 2.7	4.7 ± 2.3	91	**<0.001**	0.46
Intrahepatic lipid content_-MRS_ [%-abs.]	9.4 [5.1; 17.1]	5.3 [2.4; 10.5]	110	**<0.001**	−0.55	4.1 [1.5; 9.2]	2.5 [.7; 6.5]	92	**<0.001**	−0.50
Fasting glucose [mmol/L]	5.8 ± 0.6	5.6 ± 0.7	159	**<0.001**	0.31	5.7 ± 0.7	5.5 ± 0.8	162	**<0.001**	0.34
2 h glucose [mmol/L]	8.2 ± 1.5	7.3 ± 2.0	159	**<0.001**	0.47	8.3 ± 1.6	7.6 ± 2.0	162	**<0.001**	0.35
Fasting insulin [pmol/L]	82.0 [59.3; 115.3]	74.3 [55.5; 111.1]	165	**0.002**	−0.24	64.2 [43.2 98.0]	62.8 [44.1; 87.6]	170	**0.019**	−0.18
HOMA-IR	3.0 [2.1; 4.1]	2.7 [1.8; 3.9]	165	**<0.001**	−0.28	2.3 [1.5; 3.4]	2.0 [1.3; 3.1]	169	**0.003**	−0.23
Matsuda index	2.4 [1.7; 3.2]	2.8 [2.0;4.1]	128	**<0.001**	−0.53	2.9 [2.2; 4.6]	3.7 [2.4; 5.6]	106	**0.002**	−0.30
HIRI	38.3 [32.8; 45.5]	35.8 [31.3; 42.3]	133	**<0.001**	−0.37	34.9 [27.9; 40.9]	31.2 [25.8; 38.6]	108	**0.003**	−0.29
IGI	13.7 [8.9; 23.5]	15.2 [8.8; 19.8]	133	0.560	−0.05	8.5 [5.7; 15.7]	9.9 [6.0; 15.9]	108	0.434	−0.08
DI	32.9 [22.1; 46.1]	38.2 [22.6; 65.5]	128	**<0.001**	−0.31	28.4 [19.5; 39.2]	33.8 [23.1; 63.4]	106	**<0.001**	−0.3

Data are shown as mean ± SD (normally distributed variables) or as median [IQR] (non-normally distributed variables). Within-group differences of normally distributed variables were tested via Student’s *t*-test (one-tailed), and within-group differences of non-normally distributed parameters were tested via Wilcoxon Signed-Rank Test. *p* for within-group difference, respectively. Significant *p*-values are bolded. Effect sizes are given as Cohen’s *d* for parametric testing or Pearson’s correlation coefficient *r* for non-parametric testing. Abbreviations: IGF-1, Growth Factor 1. Insulin-like IGFBP1/-2, insulin-like growth factor binding protein-1/-2. BMI, Body mass index. WHR, waist/hip ratio. BIA, bioelectrical impedance analysis. MRI, magnetic resonance imaging. MRS, magnetic resonance spectroscopy. abs, absolute. HOMA, homeostatic model assessment. IR, insulin resistance. HIRI, hepatic insulin resistance index (Abdul-Ghani). IGI, insulinogenic index (Seltzer). DI, disposition index-2.

**Table 3 ijms-25-06400-t003:** Changes in metabolic parameters over time in association with (**a**) IGF-1 baseline levels and (**b**) IGFBP-1 baseline levels.

Parameters	Mean Difference	95% CI	*p*	*d*/*r*
(**a**)				
Subgroups of IGF-1 baseline levels: above vs. below the median				
∆ IGF-1 [µg/L]	−32.09	[−41.12; 23.05]	**<0.001**	−0.75
∆ IGFBP-1 [µg/L]	0.06	[−0.76; 0.88]	0.396 ^a^	0.05
∆ IGFBP-2 [µg/L]	17.90	[−22.16; 57.96]	0.422 ^a^	0.04
∆ Body mass index [kg/m^2^]	−0.31	[−0.68; 0.07]	0.053	−0.18
∆ Waist-to-hip ratio [cm/cm]	0.01	[−0.00; 0.03]	**0.046**	0.19
∆ Body fat content_-BIA_ [%]	0.13	[−0.74; 0.99]	0.386	0.03
∆ Visceral fat volume_-MRI_ [L]	−0.24	[−0.48; 0.00]	**0.027**	−0.28
∆ Intrahepatic lipid content_-MRS_ [%-abs.]	−1.75	[−3.44; −0.054]	**0.011** ^a^	−0.18
∆ Fasting glucose [mmol/L]	0.03	[−0.09; 0.15]	0.321	0.05
∆ 2 h glucose [mmol/L]	−0.06	[−0.46; 0.35]	0.394	−0.03
∆ Fasting insulin [pmol/L]	−11.31	[−27.74; 5.12]	**0.031** ^a^	−0.12
∆ HOMA-IR	−0.36	[−0.96; 0.25]	0.086 ^a^	−0.09
∆ Matsuda index	0.45	[−0.09; 0.98]	**0.019** ^a^	0.15
∆ HIRI	−1.16	[−3.35; 1.02]	0.232 ^a^	−0.08
∆ IGI	−5.11	[−10.84; 0.62]	**0.118** ^a^	−0.10
∆ DI	−7.59	[−22.13; 6.95]	0.679 ^a^	−0.03
(**b**)				
Subgroups of IGFBP-1 baseline levels: below vs. above the median				
∆ IGF-1 [µg/L]	15.49	[5.97; 25.00]	**<0.001**	0.34
∆ IGFBP-1 [µg/L]	1.79	[0.99; 2.58]	**<0.001** ^a^	−0.22
∆ IGFBP-2 [µg/L]	−3.60	[−43.78; 36.58]	0.430 ^a^	0.00
∆ Body mass index [kg/m^2^]	−0.17	[−0.54; 0.20]	0.183	−0.10
∆ Waist-to-hip ratio [cm/cm]	0.00	[−0.01; 0.02]	0.465	0.01
∆ Body fat content_-BIA_ [%]	−0.26	[−1.13; 0.61]	0.276	−0.07
∆ Visceral fat volume_-MRI_ [L]	−0.11	[−0.36; 0.14]	0.193	−0.13
∆ Intrahepatic lipid content_-MRS_ [%-abs.]	−1.28	[−2.95; 0.38]	**0.049** ^a^	−0.14
∆ Fasting glucose [mmol/L]	0.05	[−0.08; 0.17]	0.221	0.08
∆ 2 h glucose [mmol/L]	−0.12	[−0.53; 0.28]	0.275	−0.06
∆ Fasting insulin [pmol/L]	−6.11	[−22.58; 10.35]	0.642 ^a^	−0.03
∆ HOMA-IR	−0.22	[−0.82; 0.39]	0.703 ^a^	−0.02
∆ Matsuda index	0.27	[−0.29; 0.83]	0.484 ^a^	−0.05
∆ HIRI	−0.60	[−2.82; 1.62]	0.785 ^a^	−0.02
∆ IGI	1.72	[−3.75; 7.19]	0.375 ^a^	−0.06
∆ DI	4.07	[−9.74; 17.89]	0.786 ^a^	−0.02

Between-group differences of normally distributed variables were tested via Welch’s *t*-test (one-tailed), and between-group differences of non-normally distributed parameters were tested via Mann–Whitney U Test. ^a^ non-parametric testing. *p* for within-group difference, respectively. Significant *p*-values are bolded. Effect sizes are given as Cohen’s *d* for parametric testing or Pearson’s correlation coefficient *r* for non-parametric testing. ∆ = Delta. Abbreviations: IGF-1, growth factor 1. Insulin-like IGFBP1/-2, insulin-like growth factor binding protein-1/-2. BIA, bioelectrical impedance analysis. MRI, magnetic resonance imaging. MRS, magnetic resonance spectroscopy. abs, absolute. HOMA, homeostatic model assessment. IR, insulin resistance. HIRI, hepatic insulin resistance index (Abdul-Ghani). IGI, insulinogenic index (Seltzer). DI, disposition index-2. Significant associations for IGF-1 adjusted for age to baseline: ∆ IGF-1: *p* = <0.001; ∆ waist-to-hip ratio: *p* = 0.098; visceral fat content: *p* = 0.054; ∆ intrahepatic lipid content: *p* = 0.042; ∆ fasting insulin: *p* = 0.185; ∆ Matsuda index: *p* = 0.115.

## Data Availability

The data presented in this study are available upon request from the corresponding author. (The data are not publicly available due to privacy restrictions.)

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
