# Peer review of "IGF-1 and IGFBP-1 as Possible Predictors of Response to Lifestyle Intervention—Results from Randomized Controlled Trials"

_ijms, 2024, doi:10.3390/ijms25126400_

Round 1

Reviewer 1 Report

Comments and Suggestions for Authors

The main aim of this manuscript was to analyze if IGF-1 and IGFBP-1 markers can predict the success of lifestyle interventions, thereby possibly guiding personalized strategies for prediabetes. The authors analyzed fasting serum levels of IGF-1, IGFBP-1, and IGFBP-2 in 345 high-risk prediabetic participants, and correlated them with changes in metabolic and anthropometric parameters.

The subject of the manuscript is interesting and in the scope of the journal, bringing an important issue. The authors' consent and observations are interesting and I recommend the publication of the manuscript after minor revisions.

Comment 1: Tabe 2a and 2b are cut, please change the page orientation for the tables.

Comment 2: Are there any changes in IGF1 or IGFBP1 levels regarding the different dietary interventions?

Reviewer 2 Report

Comments and Suggestions for Authors

The authors analyze the serum levels of IGF-1, IGFBP-1 and IGFBP-2 at baseline and 1 year post intervention in 345 prediabetic participants from 3 different dietary intervention trials. They conclude that those with higher IGF-1 levels and lower IGFBP-1 levels at baseline had greater improvements in metabolic outcomes following intervention. 

Table 2 which shows the metabolic parameters in those with low vs high IGF-1 and IGFBP-1 is cut-off.  In this analysis the authors use the median to stratify low vs high levels of IGF-1 and IGFBP-1. It would be good to also perform sensitivity analyses using tertiles or quartiles. 

IGF-1 is known to decrease with age. In their analyses, the authors did not appear to have adjusted for age. It could be that those who are older and have lower IGF-1 have a weaker allostatic response and hence do not respond as well to the interventions.

In Table 2, it would be good to compare the metabolic parameters in the low vs high groups at baseline. It appears that those with higher IGF-1 had lower insulin resistance. Also those with higher IGFBP-1 had lower BMI, lower liver fat and visceral fat and lower insulin resistance. It could be that those with lower IGFBP-1 have greater improvements in response to the interventions because they have higher BMI, liver and visceral fat and higher insulin resistance at baseline. 

In Table 3 the authors show the change in metabolic parameters with time by comparing the subgroups of IGF-1 and IGFBP-1 stratified using the median. As the improvement in metabolic parameters occurs for higher IGF-1 and lower IGFBP-1, it would be good if the comparison could be for higher IGF-1 vs lower to show the negative change in metabolic parameters to be consistent. It is not indicated in the legend whether the p-values are adjusted for multiple comparisons, also whether the p-values are adjusted for change in BMI. Perhaps the authors could also adjust for baseline differences in insulin resistance and visceral fat as can be seen in Table 2 that these exist.

In lines 264-266 the authors claim that lower IGFBP-1 is associated with greater improvement in 2h glucose and IHL but in Table 3 the change in 2h glucose was not statistically significant for low IGFBP-1 compared to high IGFBP-1. Since IGF-1 and IGFBP-1 are continuous variables it would be good if they could be analyzed as such for Table 3 instead of dichotomizing to 2 groups based on median.

Round 2

Reviewer 2 Report

Comments and Suggestions for Authors

The authors have addressed all the comments adequately.